# Prospective cohort study evaluating feasibility, acceptability, and clinical impact of diabetes self-management education in a PEN-Plus program in Southeastern Liberia

Gina Ferrari[1,2][☯]*, Joe Davies[3][☯], Ada Thapa[1][☯], Cyrus Randolph[3], Marwarline Wernah[3], Howe Wodoblita[3], Theophilus T. Allison[3], Jacqueline Pierre[3], Sterman Touissant[3], Alphonso Jallah[3], Anthony Tucker[4], Joshua F. Bartue Sr.[4], Emmanuel Flomo[4], Gedeon Ngoga[2,5,6], Amy McLaughlin[2,3], Apoorva Gomber[1], Matthew M. Coates[1], Paul H. Park[1,7], Gene Bukhman[1,2,7‡], Alma J. Adler[1,7‡], Celina Trujillo[1,2,8‡]

**1** Division of Global Health Equity, Department of Medicine, Center for Integration Science, Brigham and Women's Hospital, Boston, Massachusetts, United States of America, **2** Partners In Health, Boston, Massachusetts, United States of America, **3** Partners In Health Liberia, Harper, Liberia, **4** Republic of Liberia Ministry of Health, Monrovia, Monsterrado, Liberia, **5** NCD Program, Inshuti Mu Buzima, Partners In Health, Kigali, Rwanda, **6** Non-Communicable Diseases Division, Rwanda Biomedical Center, Kigali, Rwanda, **7** Harvard Medical School Department of Global Health and Social Medicine, Program in Global Noncommunicable Disease and Social Change, Boston, Massachusetts, United States of America, **8** Department of Family Health Care Nursing, University of California, School of Nursing, San Francisco, California, United States of America

☯ The first co-authors contributed equally to this work.
‡ The senior co-authors also contributed equally to this work
* gferrari@bwh.harvard.edu

## Abstract

People living with insulin dependent diabetes (PLWIDD) in low and lower-middle income countries (LLMICs), specifically rural areas, face significant barriers to diabetes management. Diabetes Self-Management Education (DSME) is an evidence-based intervention to educate and empower people living with diabetes to improve self-management. This pilot study evaluates the feasibility and clinical impact of a DSME program in a rural Package of Essential Noncommunicable Disease-Plus (PEN-Plus) clinic in Liberia. This study was conducted at two sites in Harper, Liberia. After being trained on how to administer DSME, clinic providers delivered DSME to PLWIDD during routine monthly clinic visits. Primary outcomes included acceptability, adoption, fidelity and the frequency of self-management behaviors; secondary outcomes included knowledge change and Hemoglobin A1c (HbA1c). Three providers and twenty-six PLWIDD were enrolled. DSME was feasible and well accepted by providers and PLWIDD. Patient knowledge scores increased from 33% at baseline to 64.6% at month 12. Average weekly blood glucose checks increased from 4.0 (95% CI 1.93, 6.23) checks per week month one, to 7.2 (95% CI 3.56, 10.79) checks per week month 12. Average weekly missed insulin injections decreased from 3.9 (95% CI 2.51, 6.03) in month one, to 0.82 (95% CI -0.14, 1.79) in month 12. Mean

**Data availability statement:** Data has been uploaded as supplemental information.

**Funding:** This work was supported by the Rierson Family to GB. The funders had no role in the study design, data collection, analysis, decision to publish, or preparation of the manuscript.

**Competing interests:** The authors have declared that no competing interests exist.

HBA1C decreased from 12.0% (107.7 mmol/mol) at baseline to 10.9% (95.6 mmol/mol) month 12, (modeled mean reduction of 1.1%,95% CI 0.0 to 2.0). This study adds to the growing body of literature showing that DSME can address many barriers faced by PLWIDD in LLMICs. Further research is warranted to expand the scope of self-management education to other noncommunicable conditions in the context of integrated PEN-Plus strategies.

## Introduction

Diabetes mellitus is a chronic, non-communicable disease (NCD) with an increasing public health burden disproportionately affecting those in low resource settings. In Africa, incidence of diabetes is expected to increase by 129% to 55 million in 2045 [1]. Sustained glycemic control significantly reduces risk for early morbidity and mortality from diabetes [2], yet the majority of people living with diabetes (PLWD) do not meet globally defined targets. People living in low and lower-middle income countries (LLMICs), especially rural areas, face significant barriers to achieving glycemic goals. In rural Liberia, specific barriers include: the scarcity and high cost of medications and appropriate food, the need to travel far distances to receive care without transportation reimbursement, the lack of awareness about diabetes, and the effect of diabetes stigma in the community [3]. These challenges are often exacerbated in people living with insulin dependent diabetes (PLWIDD). Thus, it is even more essential that patients have a strong clinical understanding and problem-solving skills in this setting.

To improve care for PLWIDD living in rural areas, in 2017 Liberia initiated the PEN-Plus strategy of care for severe chronic conditions beginning with Maryland County in the impoverished Southeastern part of the country. PEN-Plus (the Package of Essential Noncommunicable Disease Interventions – Plus) is a comprehensive strategy to decentralize care for multiple severe NCDs including insulin-dependent diabetes by training and equipping mid-level providers to manage several chronic conditions using shared infrastructure [4,5]. In 2022, all 47 countries of the World Health Organization's African region adopted a resolution to initiate PEN-Plus programs by 2030 [6,7].

The goal of PEN-Plus programs in this setting is to achieve an "intermediate" level of care to reduce the risk of morbidity and mortality from diabetes. "Intermediate" care involves two to four blood glucose checks per day, multiple daily injections of insulin, availability of Hemoglobin A1C (HbA1c) testing and other complication screening, and access to diabetes education with social support [8]. PEN-Plus programs, including the program in Liberia, have access to insulin, blood glucose strips, and HbA1c testing, but may not have formalized plans for providing diabetes education, addressing the psychosocial burden of living with diabetes, or empowering and engaging patients with their own care.

Diabetes self-management education (DSME) is an evidence-based, structured method to educate and empower PLWD to improve self-care by focusing on the

behavioral aspects of diabetes management and problem-solving to address barriers to self-management [9]. The Association of Diabetes Care and Education Specialists (ADCES) defines seven key self-management behaviors: healthy eating, being active, monitoring, taking medication, problem-solving, healthy coping, and reducing risks [10,11]. The theoretical basis for DSME draws from several behavior change models, including the health belief model [12], social cognitive theory [13], and the transtheoretical model [14]. Practically, DSME involves a shift in clinic culture to center patient education and psychosocial needs. Specific tools used in DSME include use of non-judgmental language (i.e., "person with diabetes" instead of "diabetic") [15], motivational interviewing, and SMART (Specific, Measurable, Achievable, Relevant, Time-bound) goal setting.

In high resource settings, DSME has been shown to improve diabetes knowledge, self-management behaviors, clinical outcomes, and psychosocial wellbeing while also reducing cost to healthcare systems [16,17]. However, research in LLMICs remains limited and focused on type 2 diabetes (T2D). A recent scoping review based in Africa evaluated 19 studies about DSME inclusive of patients with type 1 diabetes (T1D), T2D, or both. Of the four studies that evaluated self-management behaviors, none found positive impact. However, 10 of the 14 studies evaluating change in HbA1c found positive impact [18]. These findings are supported by a systematic review by Lamptey et al, which found an improvement in HbA1c in all nine studies evaluated ranging from 0.5% to 2.6% [19]. Several randomized control trials found similar reduction in HbA1c, but did not measure behavioral or psychosocial outcomes [20,21]. To date, no studies of DSME have looked specifically at PLWIDD, feasibility outcomes, or psychosocial impact of DSME in LLMICs.

This study aims to:

1. Evaluate change in provider knowledge of DSME delivery through training and mentorship.

2. Assess the feasibility of providers delivering DSME to PLWIDD in a rural region in Liberia.

3. Measure the effect of DSME on self-management behaviors, psychosocial well-being and clinical outcomes among PLWIDD.

4. Determine acceptability of DSME among providers and PLWIDD.

## Materials and methods

### Study design

This was a pilot, prospective dual-cohort study designed to evaluate feasibility, acceptability, and effect on clinical outcomes of implementation of DSME in a rural NCD clinic. The study was initially designed in three overlapping phases described in the previously published protocol [22]. However, due to limited resources during the COVID-19 pandemic only phases one and two were completed. In June 2021, a patient advisory board (PAB) was established with patients who were identified by providers to be engaged in the clinic and interested in helping other patients. These patients volunteered to meet monthly with providers to provide feedback on changes being instituted to clinic structure as part of the study. Phase one began July 2021 and continued for 12 months. During this phase, clinicians (cohort one) who were currently providing care for PLWIDD were enrolled and received training on how to administer DSME. Phase 2 began in August 2021. In phase two, PLWIDD (cohort two) were enrolled and received structured DSME for one year administered by providers.

### Setting

The study was conducted at two health facilities in Maryland County, Liberia. Maryland County is a rural area of Liberia with a population of 172,202. JJ Dossen Memorial Hospital (JJD) in Harper City is a county referral hospital and teaching center. Pleebo Health Center (PHC) in Pleebo is a district health center affiliated with JJD. A severe NCD program was founded at both sites in 2017 using the PEN-Plus model. At both JJD and PHC PLWIDD receive refills of insulin and other

diabetes supplies during monthly consultations. Glucose data is reviewed, and insulin dosing is adjusted by a specialty trained nurse or physician assistant. The standard of care includes intermediate acting human insulin (NPH) two times daily and short acting human insulin (Regular) two to three times daily.

## Participants

Providers were invited to participate if they had received NCD training and provided care to PLWD. All PLWIDD enrolled in care at JJD or PHC in July 2021 were invited to participate in the study. Patients newly diagnosed, newly started on insulin within the two months prior to study start, or who were pregnant or became pregnant during the study were excluded. Patients excluded from analysis or who did not wish to participate in the study were allowed to participate in all education activities, but no data were collected for research purposes.

Study recruitment occurred between July 12, 2021 and August 20, 2021. Participants were informed about the study during regular clinic visits. All participants provided written informed consent. Children less than 18 years of age provided assent in addition to informed consent provided by a legal guardian. Legal guardians were present during the assent and consent process.

## DSME intervention

Three providers (one physician assistant and two nurses trained in NCD care) who regularly provide care for PLWD received fourteen hours of in-person, didactic training facilitated by a Nurse Practitioner and a NCD Nurse Trainer specializing in diabetes care and education over the course of two weeks. Training curriculum emphasized behavioral and educational aspects of diabetes management, including use of person-first language, teach-back, SMART goal setting, motivational interviewing, and the seven self-care behaviors of DSME [10,11]. Training curriculum was adapted from a course for Nurse Practitioner students entitled "Behavioral Approaches to Diabetes Across the Lifespan" at the University of California, San Francisco [23]. New training content was developed for this study due to the need to adapt to local context. Providers also received training on identifying and addressing mental health disorders commonly comorbid with diabetes such as depression and diabetes distress. After the initial training, providers participated in fortnightly one hour online continuing education with two Nurse Practitioners specialized in diabetes care and education. Topics varied between case discussions and further didactic training based on knowledge gaps identified in the initial provider assessment.

Next, the PAB and providers discussed changes to clinic structure that needed to be implemented to facilitate improved experience for patients and providers. All PLWIDD were asked to come to clinic together on the same day each month because this allowed for a team-based approach to care with mental health present, as well as exchange of experience and mutual support between patients. New monthly documentation was implemented to guide providers to ask more in-depth questions about self-management behaviors, with space to document a SMART goal to be decided by the patient at each visit and progress towards goals. Study participants completed screening every three months for diabetes distress and depression and were referred to the mental health provider in response to positive screenings. At baseline, PLWIDD completed a knowledge assessment to identify gaps in knowledge of diabetes self-management. Scores on the knowledge assessment were used to guide individualized patient education.

### Primary outcomes

**Acceptability.** Acceptability of DSME was assessed by qualitative interviews with PLWIDD and providers at baseline and one year.

**Adoption.** Adoption was assessed through NCD providers reflecting on challenges to adopting DSME through qualitative interviews conducted at baseline and one year.

**Fidelity.** Monthly chart reviews were used to assess the regularity in which SMART goals were reviewed to guide patient education as a proxy for fidelity.

**Self-management behaviors.** Self-management behaviors were assessed monthly through frequency of self-monitoring of blood glucose, administering of insulin, and frequency of bringing glucometers and logbooks to clinic visits.

### Secondary outcomes

**HbA1c.** HbA1c was measured using PTS Diagnostics *HbA1c+Now* HbA1c machines. The intended testing interval was at baseline and every three months.

**Retention of provider and patient knowledge.** We assessed patient and provider knowledge retention through tests at baseline, six months and one year.

**Severe adverse events.** Severe adverse events defined as diabetic ketoacidosis, severe hypoglycemia (patient unable to treat themselves) or hospital admissions related to diabetes were assessed through chart review monthly.

## Data collection and analysis

### Qualitative data

Semi-structured qualitative interviews were conducted by trained research staff members. Boston-based researchers completed interviews with providers in English. Liberia-based study staff members trained in qualitative interview techniques interviewed PLWIDD in Liberian English. Interview topics with providers and patients are described in detail in Table 1.

**Table 1. Interview topics.**

| Interviewee | Interview Topics | |
|---|---|---|
| **Providers** | Pre-DSME | • Confidence in DSME skills<br>• Satisfaction with training<br>• Perceived benefits/advantages of DSME<br>• Concerns about DSME<br>• Anticipated barriers for implementing DSME<br>• Anticipated facilitators for implementing DSME<br>• Anticipated acceptability of DSME by PLWIDD |
| | During DSME implementation | • Retention of DSME knowledge and skills<br>• Impact of continuing education/mentorship on DSME<br>• Utilization of DSME skills<br>• Utilization of DSME study forms<br>• Perceived benefits of DSME<br>• Changes in patient-provider interactions<br>• Facilitators for implementing DSME<br>• Perceived sustainability of DSME<br>• Perceived barriers to DSME sustainability |
| **PLWIDD** | Acceptability | • Satisfaction with diabetes care<br>• Satisfaction with DSME program<br>• Changes in patient-provider interactions<br>• Focus on patient-centered care<br>• Included in decision making<br>• Recommended changes to clinic structure |
| | Knowledge and skills | • Changes in self-management<br>• Barriers and facilitators to self-management |
| | Self-confidence | • Knowledge/understanding of diabetes<br>• Confidence in ability to self-manage |
| | Psychosocial wellbeing | • Thought/ feelings about living with diabetes<br>• Burden of diabetes management<br>• Disclosing diabetes status |

DSME-Diabetes Self-Management Education, PLWIDD: People living with Insulin Dependent Diabetes

Interviews were audio recorded and transcribed by two study staff. Interviews with PLWIDD were transcribed by a study member familiar with Liberian English and directly translated into English. A codebook was developed based on interview guides, and transcripts were independently double coded using thematic analysis. Qualitative coding and analysis were done using Dedoose [24]. Analysis utilized two iterative steps. First, we utilized an a-priori thematic analysis. Then, after the first round of coding we added additional themes that had emerged. Emerging themes were discussed between coders to increase validation. One of the two researchers who participated in the coding and analysis is based in Boston and has no affiliation with the communities or health facilities where study participants were recruited. The second researcher was involved in training providers on DSME.

## Clinical reviews

**Chart reviews.** Charts were reviewed to identify severe adverse events and to ensure regularity of SMART goal review. Logbooks and glucometers were reviewed during monthly visits to calculate the number of times blood glucose was checked in the week preceding the clinic visit. Patients reported the number of times they administered their insulin in the week preceding the clinic visit. The mean of each indicator was reported by month.

**Mentoring checklists.** A mentoring checklist was used to evaluate providers' use of DSME. Two standard clinical simulation scenarios were developed to assess growth in provider's communication skills. One was a standardized patient for a typical diabetes visit, while the other was a patient with alcohol use disorder and diabetes. Providers completed an online, synchronous "visit" with a mentor role-playing as the patient. An additional mentor observed the interaction and completed the mentoring checklist to provide feedback. Mentoring checklists were completed at months three and six. Checklist scores were tabulated for each provider and compared between each patient encounter.

## Quantitative analysis

All quantitative data analysis was completed in STATA v 15.1 and R version 4.2.2.

**HbA1c.** Due to procurement issues with HbA1c cartridges, no HbA1cs were checked at month nine. Some individuals were missing measurements for particular time points; estimates for all available data and for the subset of individuals without missing measurements are shown at any time point. To account for repeated measures, estimated changes in HbA1c calculated from baseline using a mixed effects model are shown below. HbA1c was estimated for individual i at time t with fixed effects ($\beta_{1-3}$) on time indicators (T3, T6, and T12) for the three-, six-, and 12-month measurements, a fixed intercept for baseline ($\beta_0$), and a random intercept for individuals ($\delta_i$).

$$HbA1c_{i,t} = \beta_0 + \beta_1 T3_{i,t} + \beta_2 T6_{i,t} + \beta_3 T12_{i,t} + \delta_i + \varepsilon_{it}$$

**Knowledge assessments.** A 41-question multiple choice exam was developed to assess DSME skills and knowledge of diabetes management. This assessment was designed to match pre-existing competencies for diabetes management within PEN-Plus, so to date has not been externally validated. Domains included psychosocial management, age-appropriate education, basic diabetes pathophysiology and management, glucose levels, glucose monitoring, insulin administration, insulin dosing, nutrition and diet, physical activity, and complications screening.

Patient knowledge assessments were developed in alignment with the ADCES seven self-care behaviors [9] and adapted with guidance from local providers to local context in Liberia. Because most patients have limited literacy, providers read the questions to the patients and marked their answers.

The mean and standard deviation of the scores on assessments were calculated at each time point. To account for repeated measures, score changes were estimated from baseline using a mixed effects model shown below, where Y is the knowledge score for individual i at time t, T6 and T12 are indicators for the six- month and 12-month assessments, respectively, and $\delta_i$ is a random intercept for individuals. Coefficients $\beta_{1-2}$ are fixed coefficients, and $\beta_0$ is a fixed intercept.

$$Y_{i,t} = \beta_0 + \beta_1 T6_{i,t} + \beta_2 T12_{i,t} + \delta_i + \varepsilon_{i,t}$$

### Deviations from protocol

Due to efforts and resources being diverted to the COVID-19 pandemic response and travel restrictions we were unable to complete some planned study activities. The proposed costing analysis through direct observation of day-to-day clinic operations was not possible. Some continuing education, training, and mentorship sessions had to be adapted to virtual simulations. Qualitative interviews with providers at six months were done virtually instead of in-person. We had planned to audio record PAB meetings to later analyze this data qualitatively, but this was not possible because meetings had to be done in open space areas and the quality of audio recordings was insufficient for transcription. Study protocol dictated data collection on psychosocial wellbeing using the Patient Health Questionnaire-9 (PHQ-9) and Problem Areas In Diabetes (PAID-5). While some instances were collected, data quality was poor in part due to missing data and in part due to inadvertent use of the PHQ-2, which has not been validated in Liberia. Therefore, these data are not included in this manuscript.

### Ethics

This study was approved by University of Liberia-PIRB Institutional Review Board (Assurance #FWA00004853) and Brigham and Women's Hospital IRB (#2021P001081).

## Results

### Demographics

Five providers were enrolled who participated in the initial training and were interviewed at baseline. Of these five, the three providers who regularly staff the NCD clinics at JJD and PHC were followed for the entire duration of the study. The other two providers who were enrolled at the beginning were not followed for the entirety of the study due to conflicting clinical responsibilities outside of NCD clinics

There were 26 PLWIDD enrolled in care at the two clinics at the onset of the study. All 26 PLWIDD were eligible, agreed to participate and provided informed consent. Eleven were from JJD while 15 were from PHC. Fourteen were female and 12 were male. The average age of participants was 34 years (range 17, 67), and eight (33%) were married. Level of education and employment varied across participants (Table 2). Twenty-four participants were living with T1D, while two participants had insulin-dependent T2D. Twelve participants were using NPH and Regular insulin, while 14 were using NPH only. The average diabetes duration was 4.3 years. The majority (81%) were receiving food or financial assistance (Table 2). One participant was withdrawn from the study during month three due to experiencing severe mental health challenges resulting in inability to complete study activities. One participant died in month 10.

### PAB

The PAB met monthly for the duration of the study, except December and January due to holidays. The PAB consisted of 12 members who were either PLWIDD or family members. Participants were selected by providers based on active involvement in the clinic. PAB members received transportation assistance to attend meetings but were not compensated additionally for their time. The PAB provided feedback and input on changes to clinic structure, promoted engagement from patients in the project, and feedback on patient education materials. They also developed a plan to engage in advancing peer support through structured peer meetings during clinic and accompaniment of providers during home visits. Since study completion, PAB members have remained active in leading peer support meetings during regularly scheduled clinic visits and regularly attend home visits with providers.

**Table 2. Participant sociodemographic characteristics.**

| Characteristics | n | % |
|---|---|---|
| *All participants* | 26 | |
| *Age, m (range)* | 34 (17, 67) | |
| *Facility* | | |
| JJD | 11 | 42.31% |
| PHC | 15 | 57.69% |
| *Gender* | | |
| Female | 14 | 53.85% |
| Male | 12 | 46.15% |
| *Marital status* | | |
| Married | 8 | 30.77% |
| Not Married | 18 | 69.23% |
| *Education Level* | | |
| None or primary | 8 | 30.77% |
| Primary | 5 | 19.23% |
| Secondary | 12 | 46.15% |
| College or more | 1 | 3.85% |
| *Occupation* | | |
| Student | 7 | 26.92% |
| Self-employed – farmer | 3 | 11.54% |
| Self-employed - business | 2 | 7.69% |
| Employee | 3 | 11.54% |
| Other | 11 | 42.31% |
| *Diabetes Type* | | |
| Type 1 diabetes | 24 | 92.31% |
| Type 2 diabetes | 2 | 7.69% |
| *Insulin Use* | | |
| NPH and Regular | 12 | 46.15% |
| NPH only | 14 | 53.85% |
| *Diabetes duration years, m (range)* | 4.30 (1.84, 7.22) | |
| *Receiving food or financial assistance* | | |
| Yes | 21 | 80.77% |
| No | 5 | 19.23% |

### Primary outcomes

**Acceptability.** Overall, patients and providers found DSME to be acceptable and superior to how diabetes was previously managed. Providers reflected on how receiving DSME training allowed them to gain a deeper understanding of patient's lived experiences with diabetes, changed their perspective on the day-to-day difficulties of diabetes management and gave them a greater sense of compassion for PLWIDD. One provider discussed specifically how their own practice has changed since receiving DSME training (Table 3, quote 1).

Despite it being perceived as a positive shift, it took time for PLWIDD to feel comfortable with these changes (Table 3, quote 2). Providers reported that patients felt they now had autonomy to make decisions about their care, which is unfamiliar in the traditional health system (Table 3, quote 2). PLWIDD also felt a main benefit of DSME was the shift in interactions with providers, substantiating some of the self-reflections described by providers (Table 3, quote 3). Overall study findings show that DSME had the unexpected benefit of improving the relationship between PLWIDD and their providers.

Global Public
PLOS Health

**Table 3. Qualitative interview quotes.**

| Quotes |
| --- |
| 1. We have achieved so many things. Sometimes when I am teaching the patients and they are not understanding, and they are not picking up I can get mad. But now DSME made me understand that this condition sometimes causes patients to be depressed, and I will take my own time to talk to them. And I will beg them to make sure that the message is conveyed. That understand what I am trying to teach them and for them to know the importance of why I am teaching them. So, yeah, it makes so many changes in the management in the NCD clinics mostly for the type 1 patients. (Provider interview) |
| 2. Yeah, it's something strange. They're not used to having a say in their care, like deciding for themselves. It's not in our health system for patients to decide what they want for their care, to improve their care. So this is something new that we are trying to get, include into the system to have the patient also make decision on their care, on what need to be done and how they themselves can do it. (Provider interview) |
| 3. Some of the things that I learned are good for me as a patient with diabetes is the way in which some of the providers used to talk to us. They used to talk to us harsh and they used to judge us on our condition. But what I learned in the study is that at least they give the provider the clarity that they shouldn't be judgmental on us even if we go wrong or we do the right thing or not do the right thing, they should take their time and talk to us and at least is improving now. (PLWIDD interview) |
| 4. Oh, compare the two. Alright. At least I'm improving in my treatment since I got on the program and since I have been part of the DSME, at least my improvement, my treatment has improved compared to when I was not enrolled in the clinic. I improved my treatment, it's, a lot of time I never had the knowledge, I never had nobody to talk to, talk to me about my condition. But now at least there are people encouraging me and I understand my treatment and I'm taking my treatment. (PLWIDD interview) |
| 5. It help a lot because we are different, different and we are in different, different, different places. So my experience is different from the other person's experience. So in group like that we share ideas, you, you bring your idea, how you feel, or what has been happening to you yourself. Myself I bring my experience, we share it and we can learn from each other in the group. (PLWIDD interview) |
| 6. Yeah, some barriers are on the starting. Because we are having a lot more patients coming. We need more staff to train to do DSME or diabetes management, so that we won't be having patients staying too long in clinics. We have more staff and maybe we have like two days of medical shared appointments instead of one day. So a few of our patients are coming to this appointment today and then next day we have few other patients coming. Because if they come one day, one of the barriers is the time. They stay too long, and they get tired and some of them missed out the appointment. (Provider interview) |
| 7. I think they are all important but like, the challenging part about them is that this is a new system in our health sector. It's strange, it's new to the patients. So sometimes we are telling them about SMART goals, it becomes challenging to understand it and how to implement it. But like, as we keep going about the process of relating the message to them on this, how to set SMART goals, the education process, they are picking up gradually, with them. But it's difficult sometimes to actually set these goals but with the interviews techniques that we have in giving, in making them to give them self-confidence of doing it, I think they are understanding it well. (Provider interview) |
| 8. I think DSME is sustainable because the conditions that we are talking about, we are involved with, there is no time duration of management. That's a lifetime thing. More patients will be diagnosed. Those patients that have been diagnosed will remain in care until they get old. So DSME is not just for the project, but something we are going to do continuously even after the research. We are happy that whatever we are doing is something that will continuously impact our health system, not just for the research but also the health system and the patients' care. (Provider interview) |

Many PLWIDD conveyed how DSME had led to an increase in knowledge and self-confidence in their ability to manage their condition (Table 3, quote 4). PLWIDD reported feeling empowered to actively participate in improving their health outcomes. (Table 3, quote 3).

Many participants mentioned more frequent engagement in self-management behaviors, especially SMBG and insulin injections. One participant noted how nutrition education, particularly the timing of meals with insulin doses, had helped with blood sugar levels. Some were proud of the way they were managing diabetes since the implementation of DSME, and others noticed their blood sugars were within range more often. Additionally, the opportunity to interact with other PLWIDD increased trust in the medical team. They particularly appreciated group education sessions as they facilitated peer learning (Table 3, quote 5).

**Adoption.** The implementation of DSME had challenges, particularly in training and implementation. Training related challenges included logistical challenges due to COVID-19, shortage of staff and growing patient load, scheduling and time constraints, and introduction to SMART goal setting (Table 3, quote 6). SMART goal setting was challenging to initiate with patients due to its novelty, but with more exposure over time patients started to gain a better understanding. Providers also reported that as patients become more comfortable with SMART goal setting, the benefits of DSME became noticeable (Table 3, quote 7).

Despite the challenges, providers felt DSME was sustainable (Table 3, quote 8) and provided recommendations on how to enhance the feasibility and adoption of DSME by incorporating other education strategies, including visuals. Their active involvement reflects a keen interest in the project and their commitment to integrating DSME into their future practice.

**Fidelity.** A total of 272 DSME monthly clinical implementation forms were analyzed. There were no major fluctuations by month in the number of monthly DSME forms completed. However, the frequency at which providers reviewed SMART goals from the previous visit varied from 39.1% to 100% (Fig 1).

Mentoring checklists completed by clinical mentors during patient simulations were used to assess utilization of DSME to guide patient encounters. All providers asked about patients' biggest concern for that visit. SMART goals set at the previous visit were reviewed in five out of six simulated patient encounters. Teach-back was evaluated only during the first simulation, and only one of the three providers used this method. Providers involved patients in decision-making regarding their treatment plan for all six simulated patient encounters observed.

**Frequency of self-management behaviors.** Participants brought glucometers to clinic at 87.1% of encounters and brought logbooks to clinic 59.6% of encounters. Average blood glucose checks increased from 4.0 (95% CI 1.93, 6.23) checks per week month one, to 7.2 (95% CI 3.56, 10.79) checks per week month 12 (Fig 2). Average missed insulin injections decreased from 3.9 (95% CI 2.51, 6.03) in month one, to 0.82 (95% CI -0.14, 1.79) in month 12 (Fig 2).

## Secondary outcomes

**HbA1c.** Mean HbA1c at baseline was 12.0% (107.7 mmol/mol) (n = 26, se 0.41). At three and six months, HbA1c was 9.1% (76.0 mmol/mol) (n = 19, se 0.64) and 10.9% (95.6 mmol/mol) (n = 17, se 0.40) respectively. At 12 months, HbA1c was 10.9% (95.6 mmol/mol) (n = 24, se 0.49) with a modeled mean reduction from baseline of 1.1% (95% CI 0.0 to 2.0) (Fig 3).

**Severe adverse events.** Over the course of the year there were five severe adverse events. There were three episodes of DKA that occurred during months 3, 6, and 8. One patient experienced hypoglycemia requiring assistance during month 7, and there was one participant death in month 10 from excessive alcohol intake.

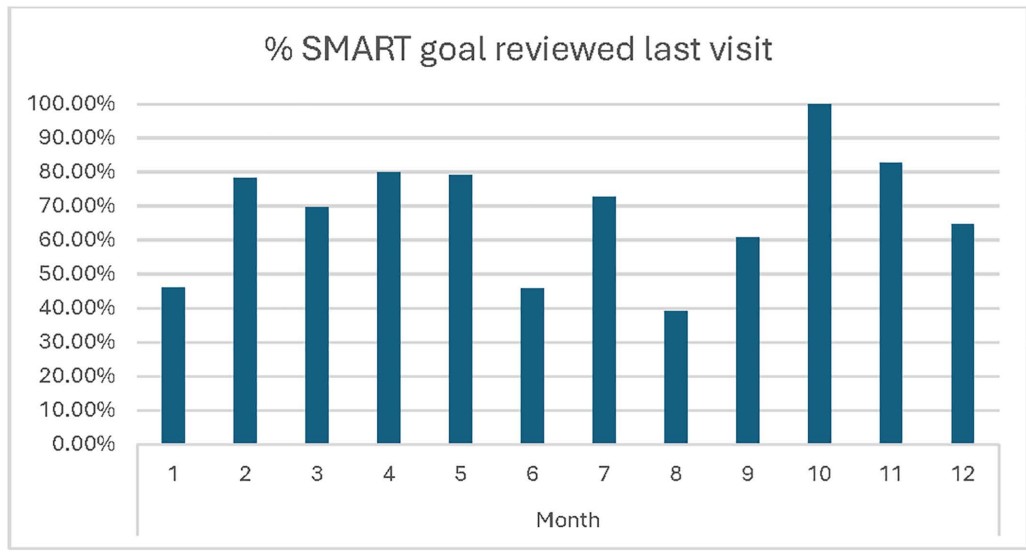

**Fig 1. Review of SMART Goal Frequency.**

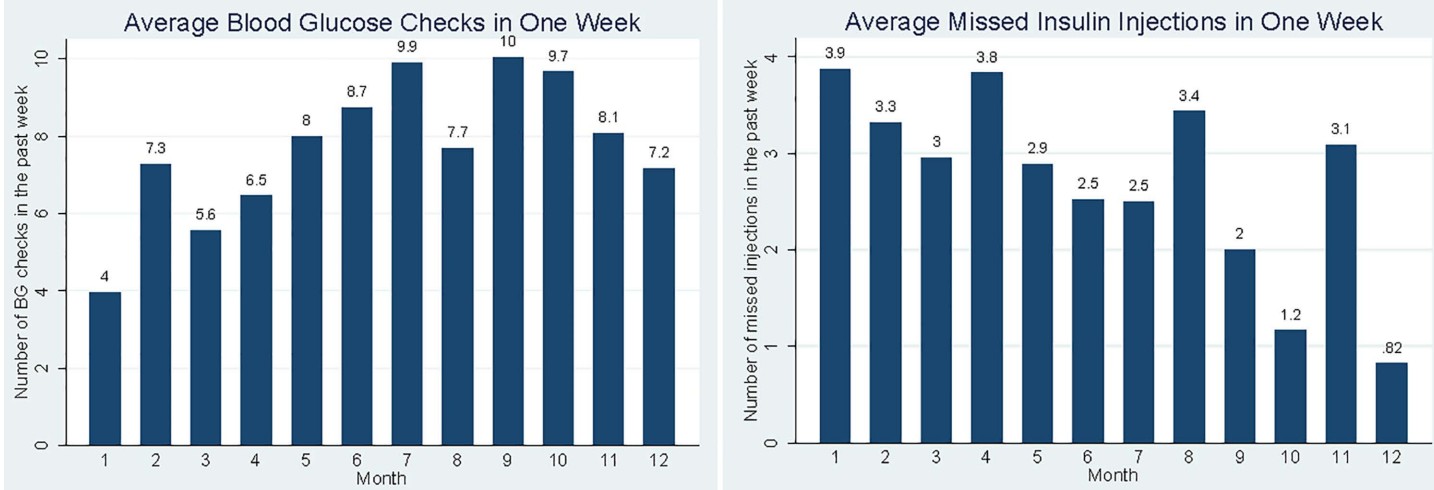

**Fig 2. Patient self-management behaviors (blood glucose checks and missed insulin injections) recorded in the week preceding clinic visits by month.**

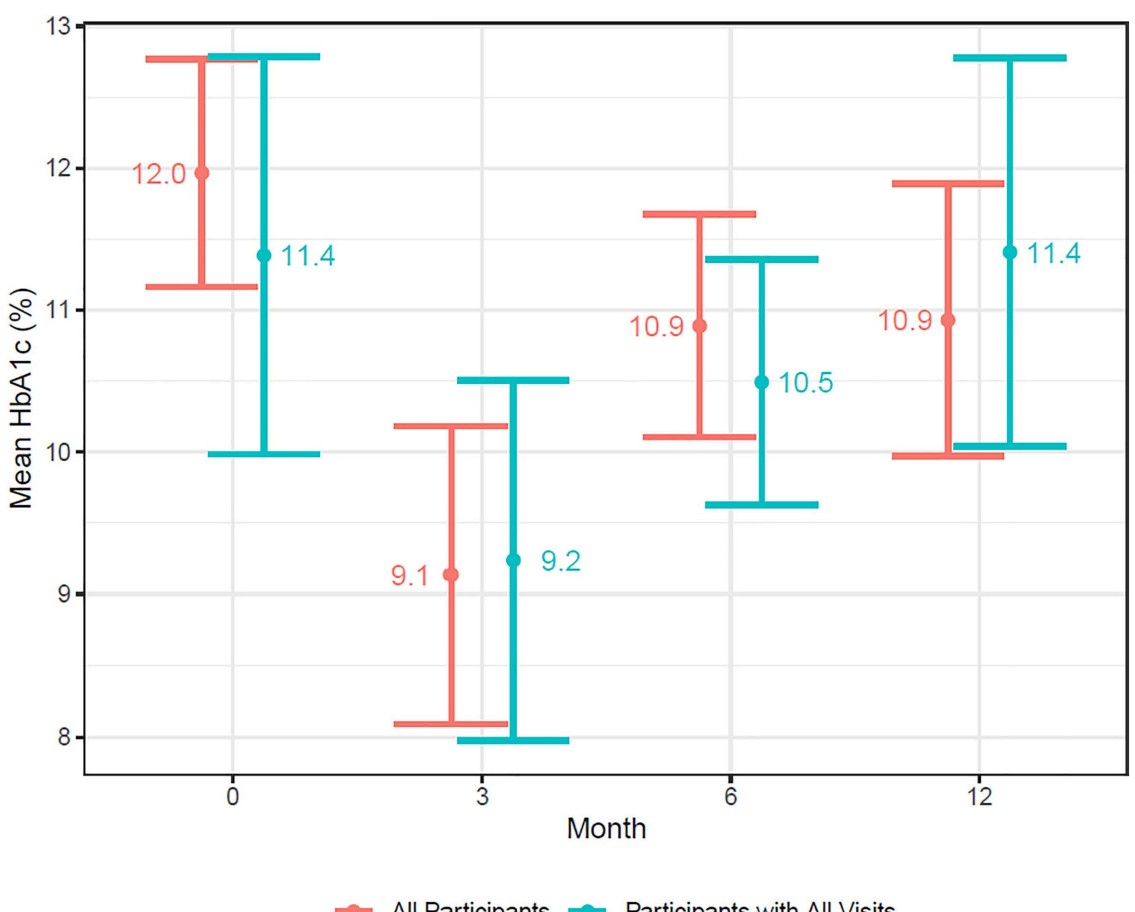

**Fig 3. Mean A1C by month, for all participants and participants who completed all study visits.**

**Provider knowledge.** Providers performed better on the knowledge assessment at six months compared to baseline. The average score increased from 60.2% (n = 3) prior to DSME training to 71.5% (n = 3) at month six. Due to travel restrictions, provider knowledge was not assessed during month 12.

**Patient knowledge.** Scores on the patient knowledge assessment increased from a mean of 32.4% (n = 26, se 4.3) at baseline to 58.1% (n = 25, se 5.7) at month six, a modeled increase from baseline of 25.0% (95% CI 15.0 to 35.1%) accounting for the slightly different composition of individuals with baseline and six-month follow-up data. At month 12, scores increased to 64.6% (n = 21, se 5.2), with an estimated mean increase from baseline of 33.7% (95% CI 23.0 to 44.3%). (Table 4)

## Discussion

This pilot study found DSME to be acceptable and feasible while having a positive impact on clinical outcomes. The intervention was supported by input from the population of PLWIDD through the establishment of a PAB. Regular PAB meetings informed implementation decisions and encouraged patient involvement and engagement in the study. Qualitative analysis suggests DSME was well accepted and useful for providers and PLWIDD alike. DSME was associated with improved patient knowledge, frequency of self-management behaviors, and HbA1c in PLWIDD. Patient knowledge assessment scores had a mean increase from baseline of 33.7% (95% CI 23.0 to 44.3%), suggesting retention of new knowledge learned throughout the DSME process. This knowledge increase was associated with an increase in SMBG and decrease in missed insulin injections. Other DSME programs in sub-Saharan Africa (SSA) examining behavioral outcomes have not shown an impact on SMBG or medication adherence, however, these studies primarily enrolled people living with type 2 diabetes and were conducted in urban areas or tertiary hospital settings [18,25,26].

Participants brought glucometers and logbooks to clinic most visits, 87.1% and 59.6% of all encounters respectively. This was less than a study conducted in a similar setting in rural Malawi, which found participants brought glucometers and logbooks to clinic 98% of visits [27]. However, the study in Malawi was specifically evaluating SMBG implementation, and more effort may have been spent reminding patients to bring supplies to clinic visits. The increases in knowledge and self-management behaviors were associated with a 1.1% decrease in HbA1c from baseline to endline. Of note, the largest observed decrease was in month three, followed by an increase in month six. This may have been the result of seasonal variability in food security, which can negatively impact diabetes management or from fatigue in adherence to protocols from providers or patients over time. Due to the small sample size, more robust studies will be needed to confirm these results. These findings do align with other studies evaluating DSME interventions in SSA, which have reported decreases in HbA1c ranging from 0.5% to 2.6% [19].

An additional noteworthy finding was the perceived cultural shift to patient-centered care. Traditionally, in Liberia, providers are seen as authority figures whose plan of care needs to be followed, without eliciting the engagement of patients. Even though it took time for PLWIDD to understand the concept of setting goals and to feel comfortable participating in the development of their care plan, results suggest that shared decision-making was feasible even in settings where patients have limited education, health literacy, and resources. Not only was this shift in patient-provider interaction notable by both providers and PLWIDD, but so was the increase in self-confidence in patients' knowledge of diabetes and ability to

**Table 4. Patient knowledge assessment scores over time.**

| | Month | N | Score (%), mean (se) | Difference from Baseline* (95% CI) |
|---|---|---|---|---|
| **All patients** | 0 | 26 | 32.4 (4.3) | |
| | 6 | 25 | 58.1 (5.7) | 25.0 (15.0 to 35.1) |
| | 12 | 21 | 64.6 (5.2) | 33.7 (23.0 to 44.3) |

*Estimated using mixed effects regression with a random intercept for each individual

manage it, increased problem-solving skills around diabetes management, and an increase in well-being and satisfaction with their care.

This study has several limitations. While the entire known population of eligible PLWIDD in Maryland county were enrolled, this totaled only 26 participants. Because there was no control group, it was not possible to show a causal relationship between the intervention and changes in outcomes. Patient knowledge assessment and psychosocial assessment were administered by providers, which may have led to participants not answering honestly. Qualitative results from the study should be interpreted cautiously due to limited literacy from some PLWIDD and risk of bias as interviews were conducted by providers who provide regular clinical care. Change in self-management behaviors were evaluated by looking at the week preceding each clinic visit, which may have over-estimated self-management behaviors. Finally, the site where the study was conducted is supported by the organization Partners In Health, which can provide medications and diabetes supplies when there are local supply chain issues. This may limit the generalizability of results to other rural clinics, where supply chain challenges can be a major barrier to diabetes care [3,28].

This study highlights several key areas for future research. Repetition of this study with a more robust design and larger sample size would be beneficial. In general, studies evaluating DSME or other behavioral interventions should consider including knowledge, behavioral, and psychosocial outcomes in addition to clinical outcomes. Though we were not able to report on psychosocial outcomes from this study due to data quality issues, we encourage inclusion of these outcomes in future research. While this study focused on diabetes, it was conducted in a clinic utilizing the PEN-Plus model, where providers also care for people living with other serious NCDs [4,29]. Other NCDs often require education and support such as healthy lifestyle counseling. Further, many of the competencies learned by providers through DSME are applicable to providing education and support to patients living with other NCDs. Further research expanding the scope of DSME to other conditions, especially in low resource settings implementing integrated care models such as PEN-Plus is warranted. Inclusion of PLWIDD themselves, whether through a PAB or peer support groups, is essential.

## Conclusion

This pilot study showed that a structured DSME program in a low resource, SSA setting was feasible, acceptable, and associated with increased patient knowledge, an improvement in self-management behaviors, and improvement in HbA1c. Further research, including expanding the scope of DSME to include other NCDs is warranted in the context of PEN-Plus expansion. As the first to evaluate DSME among PLWIDD in a rural, LLMIC setting, this study adds to the growing body of literature showing that the combination of support and education provided by DSME can address many of the barriers to care faced by PLWD globally.

## Supporting information

**S1 Data. DSME data.**
(ZIP)

**S1 Checklist. Inclusivity in Global Research questionnaire.**
(DOCX)

## Acknowledgments

Thank you to staff and colleagues that supported the study in various capacities. A special thanks to the PAB for advising us through this process and keeping patients' perspectives at the forefront, and to the Liberia MOH and PIH Liberia for allowing us to conduct this project at their facilities. We are appreciative of Carolina Noya, FNP, PhD and Maureen McGrath, RN, MS, PNP for their ongoing mentorship around teaching, training, and study design. In memory of Cyrus Randolph and his tireless commitment to helping others as a provider, mentor, colleague, and friend.

## Author contributions

**Conceptualization:** Gina Ferrari, Gene Bukhman, Alma J Adler, Celina Trujillo.

**Formal analysis:** Gina Ferrari, Matthew M Coates.

**Investigation:** Celina Trujillo.

**Methodology:** Gina Ferrari, Joe Davies, Ada Thapa, Cyrus Randolph, Marwarline Wernah, Alphonso Jallah, Anthony Tucker, Joshua F Bartue Sr, Emmanuel Flomo, Gedeon Ngoga, Amy McLaughlin, Gene Bukhman, Alma J Adler, Celina Trujillo.

**Project administration:** Gina Ferrari, Joe Davies, Cyrus Randolph, Marwarline Wernah, Howe Wodoblita, Theophilus T Allison, Jacqueline Pierre, Sterman Touissant, Alphonso Jallah, Anthony Tucker, Emmanuel Flomo, Gedeon Ngoga, Apoorva Gomber, Paul H Park, Alma J Adler.

**Supervision:** Alma J Adler, Celina Trujillo.

**Writing – original draft:** Gina Ferrari, Alma J Adler.

**Writing – review & editing:** Gina Ferrari, Joe Davies, Ada Thapa, Cyrus Randolph, Marwarline Wernah, Howe Wodoblita, Theophilus T Allison, Jacqueline Pierre, Sterman Touissant, Alphonso Jallah, Joshua F Bartue Sr, Gedeon Ngoga, Amy McLaughlin, Apoorva Gomber, Paul H Park, Gene Bukhman, Alma J Adler, Celina Trujillo.

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
