## [Decision Letter · Decision Letter 0]

2 Jun 2025

PGPH-D-25-01010

Prospective cohort study evaluating feasibility, acceptability, and clinical impact of diabetes self-management education in a PEN-Plus program in Southeastern Liberia

Dear Dr. Ferrari,

Thank you for submitting your manuscript to PLOS Global Public Health. After careful consideration, we feel that it has merit but does not fully meet PLOS Global Public Health’s publication criteria as it currently stands. Therefore, we invite you to submit a revised version of the manuscript that addresses the points raised during the review process.

We look forward to receiving your revised manuscript.

Kind regards,

Roopa Shivashankar, MD, MSc

Academic Editor

Journal Requirements:

2. In the online submission form, you indicated that [Data are available upon reasonable request. Deidentified data are available upon reasonable request from the corresponding author (GF) at gferrari@bwh.harvard.edu.].

a. In a public repository,

b. Within the manuscript itself, or

c. Uploaded as supplementary information.

Additional Editor Comments (if provided):

Reviewers' comments:

Reviewer's Responses to Questions

**Comments to the Author**

1. Does this manuscript meet PLOS Global Public Health’s publication criteria?

Reviewer #1: No

Reviewer #2: Yes

Reviewer #3: Yes

2. Has the statistical analysis been performed appropriately and rigorously?

Reviewer #1: N/A

Reviewer #2: Yes

Reviewer #3: Yes

3. Have the authors made all data underlying the findings in their manuscript fully available (please refer to the Data Availability Statement at the start of the manuscript PDF file)?

Reviewer #1: Yes

Reviewer #2: No

Reviewer #3: Yes

4. Is the manuscript presented in an intelligible fashion and written in standard English?

Reviewer #1: No

Reviewer #2: Yes

Reviewer #3: Yes

Reviewer #1: The topic of the manuscript is of interest in the LLMICs, despite the literature already available in the developed world. The concept of the study is relevant to clinical settings in LLMICs, but the execution has been less than satisfactory (probably related to the restrictions of COVID 19 pandemic) and the writing of the manuscript is voluminous for the limited work done. The number of participants is too small (3 providers and 26 persons with insulin dependent diabetes mellitus) for any meaningful conclusions to be drawn as well as for scalability and application to communities even in LLMICs. There are a few more issues that need to be addressed before this manuscript can be considered further.

I am intrigued as to how only 26 PLWIDD could represent all patients on insulin among a population of 172,202. The authors need to mention the prevalence of diabetes mellitus (both T1 & T2) in Liberia and the total number of patients on insulin in the studied area so that the readers put the study population in perspective.

The psychosocial assessment part has many inadequacies and rightly acknowledged by the authors. However, trying to present the results of that part and discussing its relevance would be misleading. Hence, it should be preferably deleted.

Specific points

Line 84 – 89

The description of PEN-Plus is confusing to the reader – PEN-Plus addresses severe NCDs like Type 1 DM, Rheumatic and congenital heart disease, sickle cell disease – the statement on lines 87 to 89 seems to suggest it is only for insulin dependent diabetes mellitus – needs to be better framed.

Line 159 to 161:

The authors state the following:

However, consistent with Essien et al in Nigeria[17], it was calculated that the study would have over 80% power using a two-tailed test with alpha value of 5% to detect a 1% decrease in HBA1c assuming a SD of 1.7 or less.

How would sample size be relevant when entire eligible population is recruited? How can sample size be calculated using one secondary outcome (HbA1c)?

Probably, this part can be deleted.

Line 167 to 178:

The DSME component needs further description – was the ‘provider’ training based on study specific modules or modules in public domain or some other source material?

Line 180

The authors state the following:

All PLWIDD were asked to come to clinic together on the same day each month.

What was the idea behind this – any specific advantage? How would this relate to up-scaling this project?

Primary Outcomes (Line 190 onwards) and Secondary Outcomes (Line 203 onwards)

The frequency/ timing of the measurement in relation to the study is mentioned for the secondary outcomes “HbA1c” and “Retention of provider and patient knowledge”. The same should be mentioned for other outcomes:

Acceptability

Adoption

Fidelity

Self-management behaviours (probably checked every clinic visit, but need to explicitly mention it)

Psychosocial wellbeing

Severe adverse events

Line 246 -247

The authors state the following:

Two standard clinical simulation scenarios were developed to assess growth in provider’s communication skills

What simulation scenarios are the authors referring to? Please describe both the scenarios.

Line 254 – 275

The mixed effects model needs clearer explanation including all symbols mentioned in the equation

Line 264 – 266

The authors state the following:

We developed a comprehensive provider assessment to assess DSME skills and knowledge of diabetes management designed to assess a preset list of competencies for diabetes management within PEN Plus.

This sentence is confusing to the reader with repetition of words like “assess” and “diabetes management”. Please simplify it and if necessary make two sentences.

With the description in the text, Table 2 is redundant and should be removed.

In each section of the results, the number of participants completing a particular visit for an outcome measure is different. Since the number of enrolled participants is small, further reduction of numbers makes the results very weak.

Line 417 – 419

For “provider knowledge”, the results are mentioned at baseline and 6 months. Please mention results at 12 months too.

There are a few grammatical/ typo errors that need to be corrected.

It is preferable to use the word “diabetes mellitus” rather than “diabetes” at least at the beginning of the manuscript.

Grammatical/typo errors/ inconsistency in language:

Few examples:

Line 48

The authors make the following statement:

People living with insulin dependent diabetes (PLWIDD) in low and lower-middle income

countries, specifically….

The abbreviation “LLMICs” should be provided here

On a similar note, the abbreviation, SSA is not expanded in the manuscript – sub-Saharan Africa

Line 139

The authors make the following statement:

Phase 2 started August 2021.

The statement “Phase 2 started August 2021” is abrupt, informal and grammatically inappropriate for a manuscript and should be better framed

Similarly, Line 162

The authors make the following statement:

Study recruitment started July 12, 2021 and completed August 20, 2021

The statement “Study recruitment started July 12, 2021 and completed August 20, 2021” –needs to be properly framed.

The word Hemoglobin A1c has been written differently in the manuscript - “HbA1c” at some places and “HBA1c” at others – please maintain uniformity.

Line 374 – 375

The authors make the following statement:

However, the frequency providers reviewed SMART goals from the previous visit varied from 39.1% to 100%

The statement should be…… However, the frequency at which providers reviewed SMART goals from the previous visit varied from 39.1% to 100%

The words in italics should be added.

Please ensure other such errors are corrected.

Reviewer #2: General Comments:

This manuscript presents a relevant and well-structured study assessing the feasibility and clinical impact of a diabetes self-management education (DSME) intervention in a resource-constrained setting. The focus on PEN-Plus implementation and involvement of a patient advisory board is commendable. However, a few clarifications and elaborations are needed to enhance the rigor and interpretability of the findings.

Specific Comments:

1. Knowledge assessment tool:

A brief description of the diabetes knowledge assessment tool should be included—specifying the number of items, the domains covered, scoring method, maximum possible score and whether the tool was validated or adapted for this context.

2. Data analysis section:

The Data Analysis section currently mentions only the software used. Please elaborate on the analytical approach adopted for key outcomes, including model specifications, handling of missing data, and any sensitivity analyses conducted.

3. Statistical modeling:

In the description of the repeated measures mixed-effect model, please specify which variables were treated as fixed effects and which were modeled as random effects.

4. Incentives for participation:

It would be helpful to clarify whether any monetary or non-monetary incentives were provided to patient advisory board members and to the providers for participating in home visits.

5. Frequency of self-management behaviors:

There appears to be a lack of internal coherence in the reporting of self-management behaviors. For example:

“Participants brought glucometers to clinic at 87.1% of encounters. The frequency stayed consistent from month one (69.2%, n=26) to month 12 (64.7%, n=12).”

“Participants brought logbooks to clinic at 59.6% of encounters. This also stayed consistent from month one (42.3%, n=26) to month 12 (47.1%, n=12).”

These statements suggest higher frequency and then month wise frequency is reported as lower. Please revise for clarity and internal consistency.

6. Referrals after PAID and PHQ screening:

The manuscript notes 26 referrals after psychosocial screening. Please clarify whether this figure refers to 26 unique patients or multiple referrals for some individuals. This information should also be included in the Results section.

Reviewer #3: This is a well-conceived, contextually relevant, and competently executed pilot study assessing the feasibility, acceptability, and preliminary impact of Diabetes Self-Management Education (DSME) among people living with insulin-dependent diabetes (PLWIDD) in rural Liberia. The manuscript addresses a critical gap in implementation science, particularly in the context of Type 1 diabetes care in low-resource settings, making it both timely and valuable. However, there are several areas where clarity and depth can be improved. Detailed comments are provided below:

• The Introduction is generally well-written. However, the description of the PEN-Plus model could be expanded for readers who may not be familiar with it.

• The study is described as a “prospective dual-cohort”, but only one cohort (PLWIDD receiving DSME) is presented. Please clarify what is meant by “dual-cohort” or revise to simply state “prospective cohort study.”

• The reference to behaviour change models (line 104) is vague. Please specify which theoretical model(s) guided the DSME intervention design.

• Clarify whether the knowledge assessment tools used were previously validated instruments or developed specifically for this study.

• Provide more detail on the Patient Advisory Board (PAB) mentioned in line 134. What was the composition of the board, and what roles did they play? Were there specific criteria for selecting patient participants?

• The lack of a comparison group limits the ability to draw causal inferences. While this limitation is acknowledged, the use of a matched pre-post design or a stepped-wedge approach could have strengthened the findings. Consider explicitly discussing this in the Discussion section.

• The observed reduction in HbA1c at 3 months (from 12.0% to 9.1%) followed by an increase at 6 and 12 months (10.9%) warrants further discussion. What might explain this trajectory, and what are the implications for intervention sustainability?

• The modelled mean HbA1c reduction of 1.1% (95% CI: 0.0 to 2.0) approaches significance but the wide confidence interval suggests uncertainty. It is important to caution readers that while the trend is encouraging, more robust data from larger samples are needed.

• The integration of qualitative quotes is effective in illustrating participant experiences. However, consider reducing the number of direct quotes in the main text to improve readability and focus.

• Where possible, report effect sizes with confidence intervals consistently (e.g., for knowledge scores, frequency of glucose monitoring) to aid interpretation.

**Do you want your identity to be public for this peer review?** For information about this choice, including consent withdrawal, please see our Privacy Policy

Reviewer #1: No

Reviewer #2: No

Reviewer #3: No

---

## [Decision Letter · Decision Letter 1]

30 Nov 2025

Prospective cohort study evaluating feasibility, acceptability, and clinical impact of diabetes self-management education in a PEN-Plus program in Southeastern Liberia

PGPH-D-25-01010R1

Dear Ms Ferrari,

We are pleased to inform you that your manuscript 'Prospective cohort study evaluating feasibility, acceptability, and clinical impact of diabetes self-management education in a PEN-Plus program in Southeastern Liberia' has been provisionally accepted for publication in PLOS Global Public Health.

Best regards,

Nicola Hawley

Academic Editor

Reviewer Comments (if any, and for reference):

Reviewer's Responses to Questions

**Comments to the Author**

Reviewer #2: All comments have been addressed

Reviewer #3: All comments have been addressed

publication criteria?

Reviewer #2: Yes

Reviewer #3: Yes

3. Has the statistical analysis been performed appropriately and rigorously?

Reviewer #2: Yes

Reviewer #3: Yes

4. Have the authors made all data underlying the findings in their manuscript fully available (please refer to the Data Availability Statement at the start of the manuscript PDF file)?

Reviewer #2: Yes

Reviewer #3: Yes

5. Is the manuscript presented in an intelligible fashion and written in standard English?

Reviewer #2: Yes

Reviewer #3: Yes

Reviewer #2: The authors have explicitly addressed the comments and adopted suggestions.

Reviewer #3: Al the comments have been addressed by the authors. However, include all the limitations in the discussion.

**Do you want your identity to be public for this peer review?** For information about this choice, including consent withdrawal, please see our Privacy Policy

Reviewer #2: No

Reviewer #3: No
